# Comparison of Properties of Poly(Lactic Acid) Composites Prepared from Different Components of Corn Straw Fiber

**DOI:** 10.3390/ijms23126746

**Published:** 2022-06-16

**Authors:** Zhongyu Qi, Baiwang Wang, Ce Sun, Minghui Yang, Xiaojian Chen, Dingyuan Zheng, Wenrui Yao, Yang Chen, Ruixiang Cheng, Yanhua Zhang

**Affiliations:** 1Engineering Research Center of Advanced Wooden Materials, Ministry of Education, Harbin 150040, China; qizhongyu@nefu.edu.cn (Z.Q.); wangbaiwang@nefu.edu.cn (B.W.); sunce@nefu.edu.cn (C.S.); yangminghui@nefu.edu.cn (M.Y.); cxj15700012290@nefu.edu.cn (X.C.); zhengdingyuan@nefu.edu.cn (D.Z.); wryao@nefu.edu.cn (W.Y.); 2021032203@nefu.edu.cn (Y.C.); 2Key Laboratory of Bio-Based Material Science and Technology, Northeast Forestry University, Ministry of Education, Harbin 150040, China

**Keywords:** corn straw fiber skin, corn straw fiber core, composite material, poly(lactic acid)

## Abstract

In recent years, under the pressure of resource shortage and white pollution, the development and utilization of biodegradable wood-plastic composites (WPC) has become one of the hot spots for scholars’ research. Here, corn straw fiber (CSF) was chosen to reinforce a poly(lactic acid) (PLA) matrix with a mass ratio of 3:7, and the CSF/PLA composites were obtained by melt mixing. The results showed that the mechanical properties of the corn straw fiber core (CSFC) and corn straw fiber skin (CSFS) loaded PLA composites were stronger than those of the CSFS/PLA composites when the particle size of CSF was low. The tensile strength and bending strength of CSFS/CSFC/PLA are 54.08 MPa and 87.24 MPa, respectively, and the elongation at break is 4.60%. After soaking for 8 hours, the water absorption of CSF/PLA composite reached saturation. When the particle size of CSF is above 80 mesh, the saturated water absorption of the material is kept below 7%, and CSF/PLA composite has good hydrophobicity, which is mainly related to the interfacial compatibility between PLA and CSF. By observing the microstructure of the cross section of the CSF/PLA composite, the research found that the smaller the particle size of CSF, the smoother the cross section of the composite and the more unified the dispersion of CSF in PLA. Therefore, exploring the composites formed by different components of CSF and PLA can not only expand the application range of PLA, but also enhance the application value of CSF in the field of composites.

## 1. Introduction

Petroleum-based plastics have flourished due to their light weight, low cost, simple process, dramatic mechanical properties and versatility, which become the raw materials for most of the current material products [1,2,3]. However, most of these petroleum-based plastics are discarded after utilization, which cannot be degraded [4], causing serious damage to the ecosystem and environment. Finally, these waste plastics form microplastics that enter the food chain, posing serious threats to human health. However, these waste plastics are costly and difficult to recycle, causing a range of environmental problems [5,6]. Over the past few decades, the global push for green and sustainable economic development has intensified, with many scholars focusing on biodegradable plastics and their composites [7,8,9,10]. Nowadays, most of the biodegradable plastics produced are polyesters, such as poly(3-hydroxybutyrate) (PHA), polymer Caprolactone (PCL), and poly(lactic acid) (PLA). These biodegradable plastics are comparable to traditional petroleum-based plastics in terms of mechanical properties and chemical properties, which could satisfy the demands for environmental protection and sustainable economic development [11,12].

Poly(lactic acid) (PLA) has been proven to be an ideal substitute for petroleum-derived polymers, and has become one of the representatives of petroleum-derived polymers, resulting from its complete biodegradability, perfect bio-compatibility, and processability. In addition, raw materials for the preparation of PLA come from renewable crops, such as corn, sugarcane, and cassava. However, the application of PLA is limited by its high cost, inherent brittleness and poor thermal stability, among other disadvantages. [13,14]. Therefore, researchers have tried to improve its inherent properties by various physical or chemical methods. Among these strategies, blending renewable filler with a PLA matrix is a simple and effective method.

Straw is one of the most abundant natural renewable materials in the world. Fibers sourced from straw have the advantages of being renewable, biodegradable and low cost. In recent years, straw fiber has received much attention as a reinforcing agent for composite materials, mainly divided into crops such as corn straw, wheat straw, rice straw and barley straw [15,16,17]. Among them, the output of corn straw fiber (CSF) is the most produced, with an annual production of around one billion tons of maize stover worldwide. However, the main disposal methods for CSF are incineration, burial and composting. These methods do not provide economic benefits and burning releases dust particles and toxic and harmful gases that ultimately affect human health [18]. In addition, in terms of industrial application, CSF is still at the primary level of processing for simple use.

Unlike other biomass fibers, CSF can be divided into the following two parts (Figure 1): corn straw fiber skin (CSFS) and corn straw fiber core (CSFC). CSFS is similar in composition to general biomass fibers, such as wood fiber and hemp fiber, and is composed of cellulose hemicellulose and lignin. CSFC is fluffy and soft, much less dense than CSFS, with strong water absorption and low mechanical properties, but strong toughness. Therefore, CSFC is suitable for use as an insulating decorative material and as a cushioning packaging material [19,20,21]. CSFS is similar to wood in main chemical composition, has poor water absorption and high mechanical strength, making it suitable for artificial boards and structural materials. CSFS accounts for ~65% and CSFC for ~35% in the CSF. Liu [22] prepared polypropylene (PP)-based wood-plastic composites (WPC) with corn straw fiber (CSF) as the raw material and found a new green way to treat agricultural waste. When the amount of crosslinking modified CSF added was 30%, the flexural strength of WPC was up to 62 MPa. Yang [23] prepared cornhusk fiber (CHF)/PP composites and researched its flexural, impact resistance, tensile, and sound absorption properties. The effect of CHF concentration, holding temperature, length of CHF, and enzyme treatments of extracted CHF on the mechanical properties of composites has been studied. A direct comparison of the others of similar construction can be used to estimate the properties of potential CHF/PP composites in the automobile interior decoration field. In view of the above, the properties of different components of CSF have been studied, which can be used as reinforcing fiber to form new composite materials with PLA. It can not only improve the disadvantages of PLA in terms of poor mechanical properties and high cost, but also improve the utilization value of CSF more efficiently, making it widely used in food packaging, construction materials and furniture [24,25,26,27,28,29].

The performance of fiber-reinforced composites in general depends on various factors, e.g., fiber size, interfacial compatibility between fiber reinforcement and the matrix, the aspect ratio of fibers and components of corn straw fiber. Due to the large polarity difference between natural fibers and PLA, the interfacial bonding between the two phases is poor and has a significant effect on the performance of composite materials, so CSF treatment is required [30,31]. Huang [32] investigated the effect of different particle sizes of wood fibers on the composites and found that the particle size of wood fiber had an obvious influence on the mechanical properties of HDPE wood-plastic composites. When the particle size of WF decreased from 10~20 mesh to 80~120 mesh, the tensile and bending modulus increased by 28.4% and 42.4%, severally. Common natural fiber reinforced polymer performance methods include solution blending, melt blending and in-situ polymerization [33,34].

Considering the physical characteristics and technological conditions of CSF and PLA, this paper uses twin-screws to melt blend CSF and PLA with different components and particle sizes to prepare CSF/PLA composites. The mechanical properties, crystallization and thermal properties of CSF/PLA composites prepared with different components were compared by means of infrared spectroscopy (FT-IR), differential scanning calorimetry (DSC), scanning electron microscopy (SEM) and thermogravimetric analysis (TG).

## 2. Results and Discussion

### 2.1. Mechanical Properties Analysis

The testing of tensile and bending properties is of great significance to the application fields and application values of composite materials. The mechanical properties analysis of the CSF/PLA composites (weight mass ratio 3:7) with different particle sizes and compositions are shown in Figure 2. PLA has the highest tensile strength and bending strength, because CSF contains a large amount of hydroxyl, which affects the compatibility of PLA and SCF. However, by changing the composition and particle size of the CSF, the decrease in mechanical properties caused by poor compatibility can be reduced. According to Figure 2a, the tensile strength and elongation at break of the SCF/PLA composites in the blended components of the CSFS/CSFC tended to increase with the decrease in the corn straw fiber particle size. When the particle size of CSF is above 120 mesh, the tensile strength and elongation at break reach the maximum value of 54.08 MPa and 4.60%, respectively. As shown in Figure 2c, when the diameter of the SCF reached more than 20–40 mesh, increases in the bending strength of all the CSFS/PLA composite samples were not significant and reached the same level of approximately 87 MPa. This is because the large dimensions of CSF are unevenly dispersed in the PLA layer and PLA cannot cover the fibers well, resulting in slippage between the CSF and the PLA matrix [35]. The CSF with large particle sizes is easy to form defects, such as voids in the interface layer of the composite material. When the sample is subjected to external force, a stress concentration point will be formed at the defect, causing it to collapse under low stress conditions. As the particle size declines, the compatibility of CSF and PLA rises. The modulus of the composites also improves. It can be observed that the tensile strength and elongation at break of the composites obtained by blending pure corn straw fibers with PLA show a tendency to increase initially and then deteriorate (Figure 2b,d). The maximum values of 80–120 mesh CSFS/PLA composites are 49.51 MPa and 4.09%, respectively, and the bending strength is 85.39 MPa. The density of CSFC is relatively low and its internal structure is dispersed, loose and porous, which cannot be used as a skeleton for support in the material, resulting in relatively poor mechanical properties [36]. In addition, the unique hollow structure of the inner core of CSFS better enhances the dispersion and compatibility of the molten PLA with the straw fibers, resulting from the skeletal support role of CSFS [19]. The interfacial compatibility of the two blended components (CSFS/CSFC) steadily improved, as was the case for the tensile and flexural strengths.

### 2.2. FT-IR Analysis

In order to analyze the chemical functional groups of the different PLA composites, the FT-IR curves of PLA before and after mixing with different components of CSF are shown in Figure 3. The CSFS has the typical stretching vibration broadband of CSF at 3350 cm^−1^, attributed to the -OH group. The absorption peak at 2822–2940 cm^−1^ was the typical stretching vibration peak of -CH (methyl and methylene groups) on cellulose and hemicellulose in CSF, and the absorption peak at 1742 cm^−1^ was attributed to the stretching vibration peaks of the ester carbonyl C=O in the CSF hemicellulose and the -CH deformation peaks (symmetric and asymmetric bending peaks) at 1385 and 1368 cm^−1^. The absorption bands at 1080 and 863 cm^−1^ were assigned to the stretching vibration of C–O–C and C–C, respectively [37]. From the FTIR spectrum of pure PLA, the asymmetric stretching vibration peak of -CH and C=O at 2987 cm^−1^ and 1752 cm^−1^ can be observed. The absorption peaks of -CH_3_ at 1353 cm^−1^ and 1071 cm^−1^ are attributed to the C–O–C stretching vibration. The absorption peak at 3350 cm^−1^ was significantly weakened after the straw fiber was blended with PLA, indicating the formation of a hydrogen bond between -OH of straw fiber and C=O of PLA [38]. The strength of the -CH stretching vibration peak generally reflects the size of the alkane chain. It can be witnessed that the -CH peak at 2987 cm^−1^ of the straw fiber/PLA composite weakens significantly compared with that of the PLA, indicating that chain scission and degradation occurred during the preparation of the composites [39]. The carbonyl stretching vibration peak at 1752 cm^−1^ decreased, indicating that the molecular chain scission and degradation occurred at the ester group of the PLA.

### 2.3. Contact Angle Analysis

The contact angle of CSF/PLA composites were determined, as shown in Figure 4. CSF contains a large number of hydroxyl groups, which are hydrophilic, so the contact angles of CSF/PLA composites were lower compared with that of PLA. However, similar to the mechanical properties, the difference between the contact angles can be reduced by adjusting the composition and particle size of CSF. The contact angles of CSF/PLA composites with particle sizes below 80 mesh are relatively small, all of which are kept below 40°. This shows that their hydrophobicity is poor. On the one hand, it can be known from the infrared spectrum that there are a huge number of hydrophilic groups on the straw fiber (Figure 4); On the other hand, composites with large particle size straw fibers have poor interfacial compatibility, with a large number of pores in the material and the aggregation of CSF leading to a decrease in the contact angle (Figure 8) [40]. With the decrease in CSF particle size, the contact angle of CSF/PLA composite increased significantly, and the contact angle of >120 CSFS/PLA composite reached the maximum of 65.48°. Interestingly, compared with that of the CSF/PLA composites, the contact angle of CSF components without CSFC is significantly higher. This can be explained by the fact that the microporous structure may give the CSFC fibers a higher specific surface area, while the contact between the water molecules and the hydrophilic groups of the CSF will worsen the hydrophobicity of the material. The contact angle of CSFC/PLA composites is only 26.74° (Figure 4i), which also proves the above viewpoint.

### 2.4. Water Absorption Analysis

The water absorption rate of the CSF/PLA composites with different components and particle sizes changes, as shown in Figure 5. According to Ilyas [41], water absorption is affected by the exposed surface area, exposure time, diffusivity, temperature, fiber content, permeability, orientation, and surface protection. In this test, the effects of different exposure time in distilled water on water absorption rate were studied at the same temperature. Due to the poor hydrophilicity of PLA itself, its water absorption is low; however, the addition of CSF significantly improves the water absorption of the composite due to the strong water absorption of the fibers. Figure 5a,b show the changes in water absorption of PLA/CSF with different particle sizes within 18 days. The water absorption of the CSFS/PLA composites gradually increased with the extension of time, and the water absorption rate reached saturation and no significant change occurred when the soaking time reached 8 days (Figure 5b). At the same time, the water absorption rate of the composite material decreases gradually with the decrease in particle size. Water in WPC can exist in three regions, including the lumen of cells, cell walls, and gaps between the filler and matrix in the interface region [42]. The process is mainly physical adsorption. While the surface of CSF/PLA composites absorbs water molecules, the cells become saturated, and then water molecules enter the interior through gaps between the filler and matrix. The large particle size CSF is difficult to be encapsulated completely by PLA, so the exposed hydrophilic groups in the fibers are more likely to bind to water molecules. At the same time, the large particle size of CSF creates pores and gaps in the composite, allowing water molecules to easily enter the interior of the composite, which in turn leads to an increase in water absorption. Reducing the exposed area of CSF, and decreasing the gap between the matrix and filler to limit water infiltration into the gap can reduce the water absorption of the material. As shown in Figure 5b, CSFS/CSFCP/PLA composites with CSFC additions have significantly improved water absorption properties due to the loose and porous structure of the CSFC component. Figure 5c,d show the saturated water absorption of CSF/PLA with different particle sizes. The saturated water absorption decreases with the decrease in the particle size of the CSF, and the lowest values are 6.51% and 6.14%, respectively. The saturated water absorption of CSFC/PLA composites can reach up to 8.35%.

### 2.5. Thermal Stability of the CSF/PLA Composites

The TG and DTG curves of the CSF/PLA composites are shown in Figure 6. As shown in Figure 6a, the decomposition process of pure PLA is between 300 °C and 370 °C, and its mass loss rate reaches 99.15%. At this stage, the molecular chain backbone of PLA is broken to produce a large amount of lactide, which is also accompanied by a small amount of carbon dioxide, carbon monoxide, acetaldehyde and other small molecular products. These products are expelled with the gas flow. No significant mass changes in the CSF/PLA composites occurred when heated to 250 °C, which were mainly due to the volatilization and evaporation of incompletely dried water molecules and other impurities in the system. All samples showed a single degradation peak, a phase attributed to the decomposition of lignin, cellulose and hemicellulose. When the temperature rises to 500 °C, the CSF begins to carbonize. When the temperature reaches 650 °C, the mass loss of the composite material is 88–92%. Among them, the smaller the particle diameter of CSF, the higher the carbon residual content. This is because the hydroxyl groups of the CSF are easily dehydrated into charcoal. The more complete the char-forming reaction of the smaller sized CSF, the greater the residual mass. It can be observed from Figure 6b,d, compared with pure PLA, that the CSF/PLA composites decreased the thermal decomposition temperature corresponding to the maximum mass loss rate of the samples from 351 °C to 292 °C, which is due to the fact that the CSF tends to produce a variety of low molecular weight inorganic substance during decomposition. It is worth noting that the thermal decomposition process of CSF/PLA composites is divided into two stages, representing the thermal decomposition process of PLA and CSF [43]. However, these two maximum mass loss rate peaks were different from those of PLA or CSF alone. The temperatures for the maximum mass loss rate of sole CSF and PLA were 305 °C and 351 °C, while that of CSF and PLA in the CSF/PLA composites were 292 °C and 304 °C. This indicates that the CSF/PLA composites are not two independent decomposition processes of CSF and PLA. It can be found that the interactions between CSF and PLA affected the pyrolysis process of composites and a synergistic effect occurred by the temperature of maximum mass loss rate [44]. Among them, phytic acid can be used as a catalyst to accelerate the degradation of PLA. The results also show that the high-temperature melting technique is suitable for the processing of CSF/PLA composites, due to its lower energy consumption for thermal decomposition [45].

### 2.6. Analysis of the Differential Scanning Calorimetry

Crystallization of the diverse CSF/PLA composites were examined by a differential scanning calorimeter (DSC). The secondary heating curve is shown in Figure 7. Table 1 shows the different values associated with the DSC curve. With the decrease in CSF particle size, the cold crystallization enthalpy of the composites increases at first and then decreases. As is known to all, the cold crystallization peak is caused by the fact that the molecular chain segment loses its ability to move when the temperature drops too fast in the molten state, and it is too late to crystallize to form an ordered structure. Therefore, the lower the enthalpy of cold crystallization, the higher the crystallinity of the material [46]. With the decrease in the particle size of the CSF, the cold crystallization temperature (*T*_cc_) of the CSF/PLA composites shifts to the low temperature region, which indicates that the crystallization ability of the material is improved. As shown in Table 1, due to the nucleation effect of CSF, the crystallization of the PLA/CSF composites is improved compared with pure PLA. The increase in crystallinity can enhance the modulus of PLA composites. Therefore, the modulus of the CSF/PLA composites also emerged as a comparable trend [47]. The crystallinity of the composites first decreases and then increases to the highest level as the CSF particle size decreases. This may be due to the fact that the compatibility of the CSF/PLA composites increases as the CSF particle size decreases, leading to an increase in the interfacial bonding of the composites, which impedes the movement of some of the PLA molecular chains and ultimately reduces the crystallinity of the PLA. When the particle sizes of CSF are lower than 120 meshes, the heterogeneous nucleation of PLA is promoted due to the reduction in CSF particle size. Furthermore, the crystallinity of the CSF/PLA composite has a tendency to decrease. At the same time, with the decrease in the particle size of the CSF, the position of the melting peak appeared to move towards the low temperature. The enhanced interfacial bonding limits the movement of PLA molecular chains and restricts crystalline growth, resulting in smaller grain sizes and thinner wafer layers.

### 2.7. Fracture Surface of the CSF/PLA Composite

The SEM images of the tensile fracture surface of the composites are shown in Figure 8. The addition of CSF will limit the mobility of the PLA molecular chains due to its larger particle size and rough, porous surface compared to PLA molecules, while its high aspect ratio fiber structure combined with PLA will greatly increase the mechanical properties of the composite. When CSF and PLA are mixed, the interfacial bonding force between the two phases is weak, and the phenomenon of rough section and holes will appear. The composites prepared with 20–40 mesh CSF as filler exhibited a inhomogeneous distribution (Figure 8a,e), with agglomerates consisting of CSFS/CSFC exposed (indicated by red circles). In addition, the relatively large number of pores and agglomerates weakens the material’s weaker tensile strength and bending modulus [48]. As shown in Figure 8d,h, as the CSF particle size decreases, the pores’, agglomerates’ as well as fibers’ exposure of the PLA/CSF composite gradually disappear. The PLA melt can better encapsulate the CSF, indicating good dispersion of the CSF in the PLA matrix, which improves the interfacial compatibility of the two phases, and thus the mechanical properties of the composite. According to the scanning electron microscope, the PLA matrix effectively wraps the exposed CSFC; the cross section of the CSFC/PLA composite is the smoothest. As the PLA melt can be filled into the cellular structure of the CSFC, it gives the composite a smooth and homogeneous cross-section. However, due to the weak rigidity of CSFC, filling it with PLA results in a significant reduction in its mechanical strength. In summary, the hybrid composition of CSFS and CSFC not only provides a homogeneous distribution and improves the interfacial adhesion of the PLA matrix composite, but also improves the mechanical properties of the composite [36].

## 3. Conclusions

In studying the feasibility of corn stalk fibers as the reinforcement of CSF/PLA composites, the effect of fraction and fiber size on the mechanical properties of the resultant composites was investigated. The particle size of straw directly affects the tensile and bending properties of the composites. The 80–120 mesh SCFS/PLA composites’ tensile strength is 49.51 MPa and bending strength is 85.39 MPa. As observed by scanning electron microscopy, the fracture morphology was smoother due to the uniform distribution of the fibers. Due to the pore structure of CSFC, the composites containing CSFC components can greatly enhance the compatibility between PLA and CSF. The contact angle test and the water absorption test show that the CSF/PLA composites with a particle size of more than 80 mesh have better hydrophobicity. This result offers a new solution for the renewable utilization of corn straw fibers and improves their economic benefits and utilization value. At the same time, this paper could provide new insights into the use of corn straw for PLA matrix reinforcement.

## 4. Materials and Method

### 4.1. Materials

PLA, grade 4032D, NatureWorks Co., Ltd. (Minnetonka, MN, USA); corn straw fiber, self-made; 20–120 mesh CSF.

### 4.2. Preparation of CSF/PLA Composites

Corn straw fiber was pulverized with the multi-functional pulverizer (800Y, Yongkang Zhaoshen Electric Co., Ltd., Yongkang, China). The CSF were separated by sieves with different mesh numbers. The CSF with particle sizes larger than 40 mesh was air-selected by the blower to separate CSFS and CSFC. Due to the differences in density, CSFC was blown away and collected under the action of the blower, while CSFS was settled due to gravity. Then, the CSFS with different meshes was pulverized with the multi-functional pulverizer. CSF, CSFS and CSFC with different particle sizes were screened by sieve and grouped as shown in the Table 2.

PLA and screened CSF of different groups were mixed evenly according to the mass ratio of 3:7. The uniformly mixed raw materials were placed into the twin-screw extruder (SHJ-20, Nanjing Jiente Machinery Co., Ltd., Nanjing, China) to obtain CSF/PLA composites (the temperature of each section was set, the first zone: 175 °C, the second zone: 180°C, the third zone: 175 °C, the fourth zone: 175 °C and the fifth zone: 170). By the micro injection molding machine (WZS10D, Shanghai Hongli Machinery Co., Ltd., Shanghai, China), the dumbbell-shaped composite sample was obtained by injection molding. The injection temperature was 190 °C, the injection molding pressure was 0.8 MPa, and the heating time was 400 s. The appearances of different PLA composites are shown in Figure 9.

### 4.3. Mechanical Property

The CSF/PLA composites’ physical and mechanical properties were measured by a testing machine (Shenzhen SANS Test Machine Co., Ltd., Shenzhen, China). According to the national standards (GB1040.3-2006), the tensile speed is 6 mm/min and the span is 40 mm. The bending test was performed at three-point bending head down speed of 6 mm/min and span 45 mm (five samples for same component were tested to calculate its average value).

### 4.4. Thermogravimetric Analysis (TGA)

The thermogravimetric analysis (NETZSCH TG 209 F3, Selb, Bavaria, Germany) was performed on 2~5 mg of the CSF/PLA composites under an argon atmosphere. The sample was conducted from 30 °C to 700 °C, with a heating rate of 10 °C·min^−1^.

### 4.5. Differential Scanning Calorimetry (DSC)

A differential scanning calorimeter (Diamod, Perkin-Elmer, USA) was used to examine the thermal properties of the CSF/PLA composites. The sample was conducted in the range of 25~200 °C at a heating rate of 10 °C·min^−1^, held for 5 min to eliminate thermal history, and then cooled at a rate of 10 °C·min^−1^ to −20 °C. The sample then was reheated to 210 °C at 10 °C·min^−1^ as the second cycle to identify the impact of CSF on crystallization and thermal behavior of composites. The crystallinity of CSF/PLA composites is calculated by Formula (1), which is as follows:(1)$$X_{c}\left(\%\right)=\frac{\Delta H_{m}-\Delta H_{cc}}{\omega \times \Delta H_{m0}}\times 100$$
where *ΔH*_cc_ is the enthalpy of cold crystallization; *ΔH*_m_ is thev; *ΔH*_m0_ is the melting enthalpy when the PLA is completely crystallized, which is 93.7 J·g^−1^; *ω* is the percentage of polylactic acid in the composite material [49].

### 4.6. Scanning Electron Microscopy (SEM)

A scanning electron microscope (QUANTA 220, FEI, Hillsboro, OR, USA) was used to observe the morphology of the fracture section of the CSF/PLA composites, the accelerating voltage was 10 kV. The samples were soaked in liquid nitrogen for more than 15 min and then after being fractured, the fractured surfaces were sputter-coated with gold before imagining.

### 4.7. Fourier Transform Infrared (FTIR)

The chemical functional groups of CSF and PLA before and after compounding with each other were examined by Fourier transform infrared spectroscopy (OPUS 7.5, Bruker). The FTIR spectra of the samples were conducted between the range of 500 and 4000 cm^−1^.

### 4.8. Water Absorption Performance

A water absorption test was performed on CSF/PLA composites with 20 × 10 × 4 mm dimensions. Each sample was dried at 30 °C for 72 h. The weight of the samples before and after immersion in distilled water for 24 h were recorded at 25 °C. Five samples for the same component were weighed to calculate its average value. The formula for calculating the water absorption rate of the sample is shown in Formula (2), which is as follows:(2)$$W=\frac{m_{2}-m_{1}}{m_{1}}\times 100$$
where *W* is the water absorption percentage (%); *m*_1_ (g) and *m*_2_ represent the quality of the sample before and after immersion, respectively [50].

### 4.9. Hydrophilic Performance

Five relatively flat sites were selected for each CSF/PLA composite, and each site was tested for hydrophilic contact with a contact angle meter. The hydrophilic contact angle at the 30 s was taken as the test value. The average value is the hydrophilic contact angle test value of the material.

## Figures and Tables

**Figure 1 ijms-23-06746-f001:**
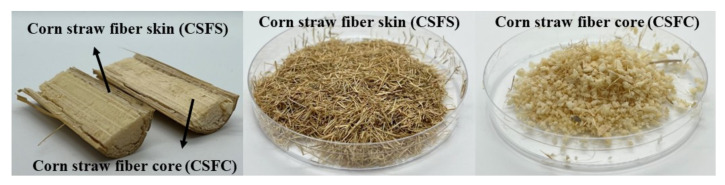
The appearance of CSF, CSFS and CSFC.

**Figure 2 ijms-23-06746-f002:**
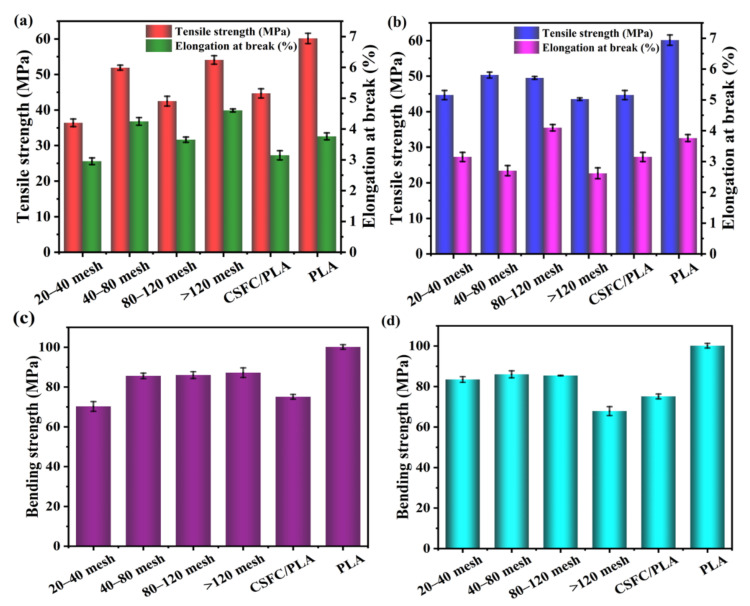
Tensile strength and elongation at break of CSFS/CSFC/PLA composites (**a**) and CSFS/PLA composites (**b**) with different particle sizes; bending strength of CSFS/CSFC/PLA composites (**c**) and CSFS/PLA composites (**d**) with different particle sizes.

**Figure 3 ijms-23-06746-f003:**
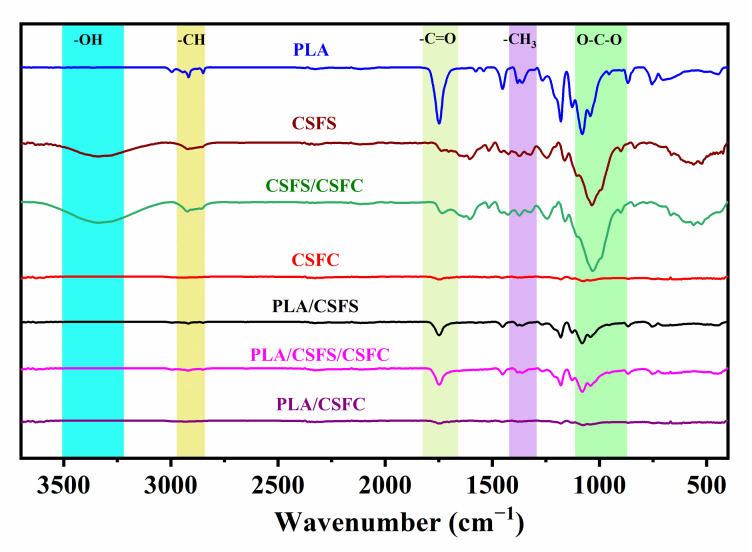
FT-IR spectra of PLA before and after mixing with CSF with different components.

**Figure 4 ijms-23-06746-f004:**
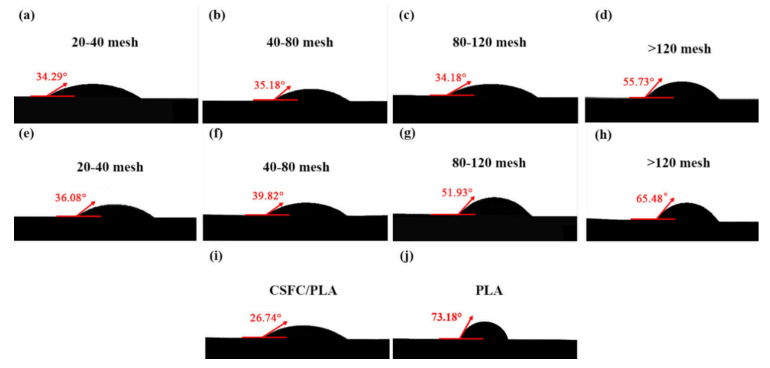
The contact angle change picture of CSFS/CSFC/PLA composites (**a**–**d**), CSFS/PLA composites (**e**–**h**), CSFC/PLA composites (**i**) with different particle sizes and PLA sample (**j**).

**Figure 5 ijms-23-06746-f005:**
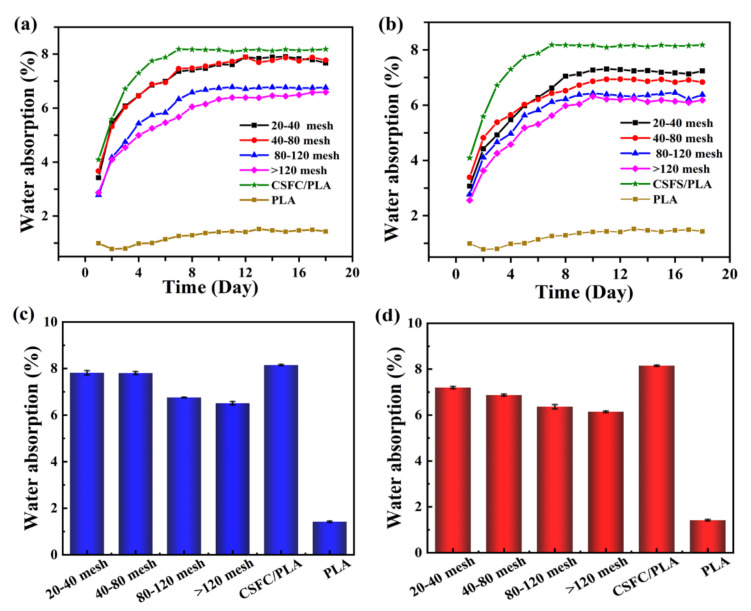
Changes in water absorption curves of CSFS/CSFC/PLA composites (**a**) and CSFS/PLA composites (**b**) within 18 days with different particle sizes; saturated water absorption of CSFS/CSFC/PLA composites (**c**) and CSFS/PLA composites (**d**) with different particle sizes.

**Figure 6 ijms-23-06746-f006:**
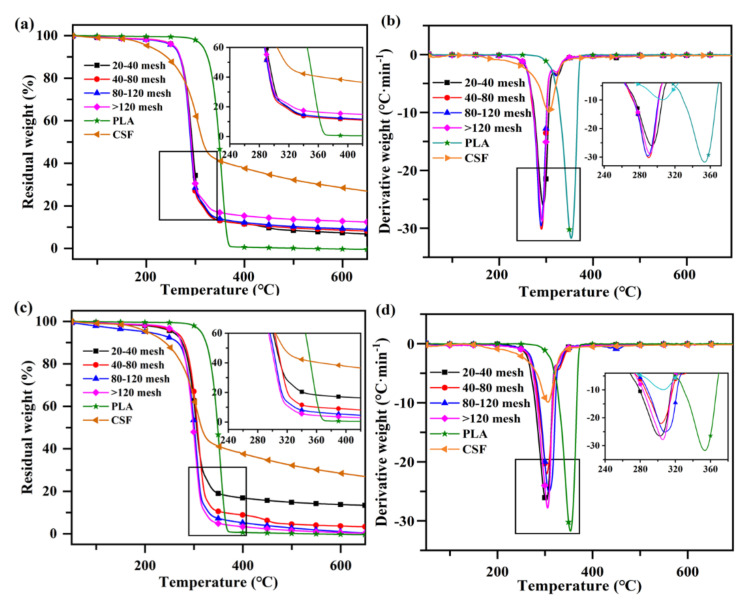
TGA and DTG curves of CSFS/CSFC/PLA composites (**a**,**b**) and CSFS/PLA composites (**c**,**d**) with different particle sizes.

**Figure 7 ijms-23-06746-f007:**
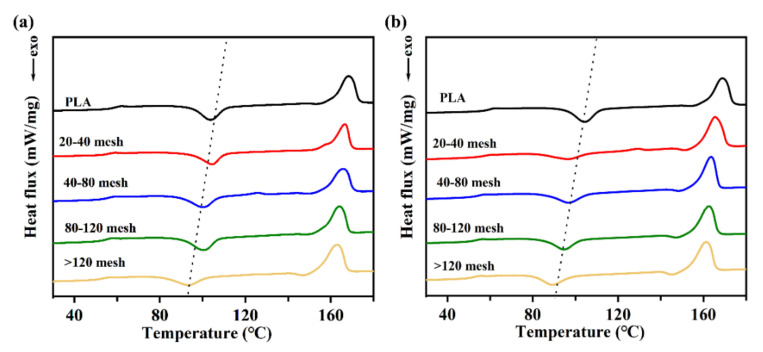
DSC curves of CSFS/CSFC/PLA composites (**a**) and CSFS/PLA composites (**b**) with different particle sizes.

**Figure 8 ijms-23-06746-f008:**
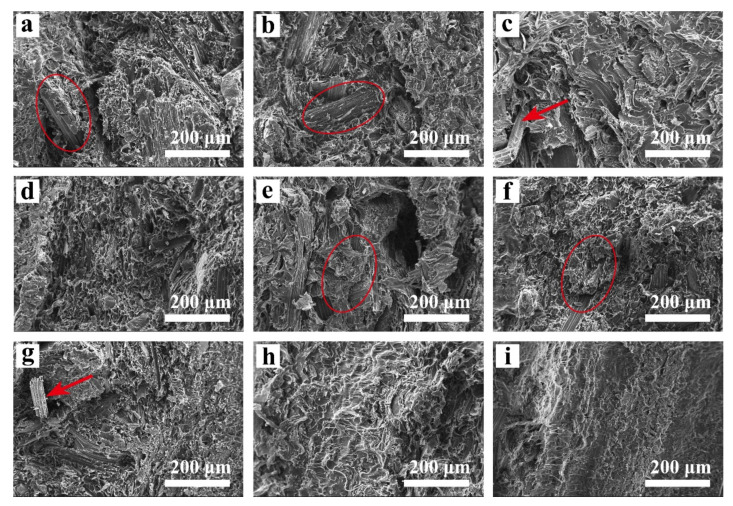
SEM images for the fracture section of CSF/PLA composites with different particle sizes: (**a**) 40 mesh CSFS/CSFC/PLA, (**b**) 40–80 mesh CSFS/CSFC/PLA, (**c**) 80–120 mesh CSFS/CSFC/PLA, (**d**) >120 mesh CSFS/CSFC/PLA, (**e**) 40 mesh CSFS/PLA, (**f**) 40–80 mesh CSFS/PLA, (**g**) 80–120 mesh CSFS/PLA, (**h**) >120 mesh CSFS/PLA, (**i**) CSFC/PLA.

**Figure 9 ijms-23-06746-f009:**
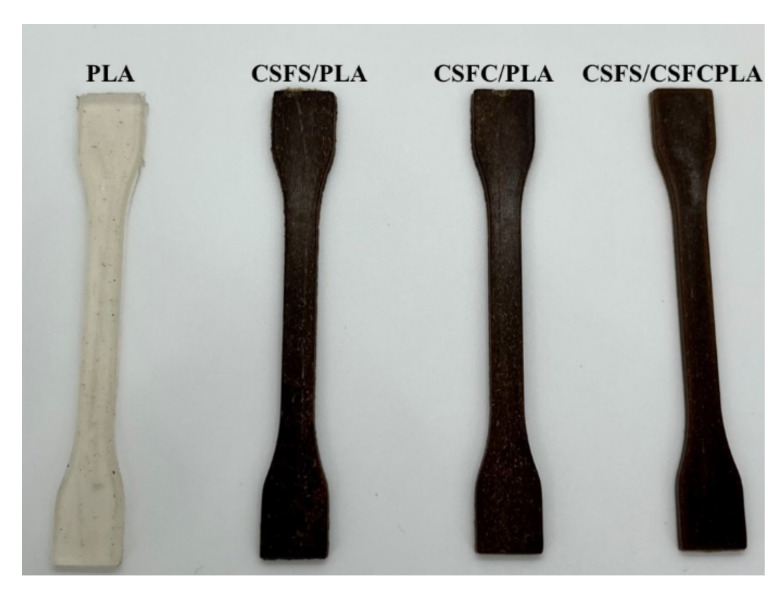
The appearance of different PLA composites.

**Table 1 ijms-23-06746-t001:** Characteristic DSC parameters of CSFS/CSFC/PLA composites and CSFS/PLA composites with different particle sizes.

Sample	*T*_g_/°C	*T*_cc_/°C	*T*_m1_/°C	∆*H*_cc_/(J·g^−1^)	∆*H*_m_/(J·g^−1^)	*X*_c_/%
PLA	59.08	104.04	169.01	13.60	29.80	24.70
CSFS/CSFC 40 mesh	55.26	104.34	167.10	9.20	26.90	26.99
CSFS/CSFC 40–80 mesh	54.54	100.14	166.28	10.80	27.30	25.16
CSFS/CSFC 80–120 mesh	53.56	100.04	164.24	13.10	28.50	23.48
CSFS/CSFC >120 mesh	52.67	94.60	163.25	9.60	30.60	32.02
CSFS 40 mesh	56.67	99.16	165.46	5.30	28.70	35.68
CSFS 40–80 mesh	53.62	97.21	163.44	9.60	26.60	25.92
CSFS 80–120 mesh	52.26	94.57	162.29	9.90	27.30	26.53
CSFS >120 mesh	51.49	89.97	161.73	8.60	26.80	27.75

**Table 2 ijms-23-06746-t002:** CSF with different components and particle sizes.

Sample	CSF (Mesh)	CSFS (Mesh)	CSFC (Mesh)
1	20–40	-	-
2	40–80	-	-
3	80–120	-	-
4	>120	-	-
5	-	40	-
6	-	40–80	-
7	-	80–120	-
8	-	>120	-
9	-	-	40–120

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
