# Peer review of "Comparison of Properties of Poly(Lactic Acid) Composites Prepared from Different Components of Corn Straw Fiber"

_ijms, 2022, doi:10.3390/ijms23126746_

Round 1

Reviewer 1 Report

This article describes the preparation and properties of poly(lactic acid) (PLA) composites with corn straw fibers (CSFs). A series of the composites were systematically prepared. The CSFS/CSFC/PLA composite with small particle size was found to show significantly improved tensile properties. The CSF/PLA composites with small particle size also showed relatively high contact angle with water indicating their higher hydrophobicity. These findings should contribute to the development of renewable PLA-based composite materials. I think that this article should be accepted for the publication in International Journal of Molecular Sciences.

1) I recommend the authors to discuss the differences and similarities between the PLA composites with the present CSFs and those with other fibers.

2) I suggest the authors to show the pictures of CSFS, CSFC, and some composites with PLA.

3) If “corn stover fiber” is the same with “corn straw fiber”, the notation should be unified to avoid readers’ confusion.

4) Figure 1: Please define what is “core”. If it means CSFC/PLA composite, it should be revised to “CSFC/PLA”.

5) I propose to compare the tensile properties, contact angles, and water absorption data of the CSF/PLA composites with those of pure PLA.

6) The authors mentioned, “At this stage, the molecular chain backbone of PLA is broken to produce carbon dioxide, acetaldehyde and other small organic compounds, which are expelled with the gas flow. ” (page 8, line 276-278). However, the major product from the thermal decomposition of PLA is lactide in general.

7) The authors mentioned, “However, the crystallinity of the composites decreases as the CSF particle size decreases.” (page 9, line 309-310). However, CSFS/CSFC >120 mesh showed the highest crystallinity.

8) This manuscript contains several careless mistakes. The authors should check it again carefully.

Page 6, line 234: “Fig. 7” ---> “Fig. 3”

Reviewer 2 Report

The topic approached in the paper and the experiment performed are of interest and present novelty, respectively potential for the development of knowledge in the field of paper.

However, the paper requires major improvements. The paper is not written in an academic way, it is not sufficiently documented, the research methods are not presented. The paper is written for the national public and needs to be improved in order to be understood internationally.

I recommend the following major improvements:

1. Documentation of the paper, the references presented are insufficient to present the situation of using corn straw fiber in WPC.

2. In the chapter on materials, these are not presented. The methods used show the national standard number. These are known nationally, it is recommended to present the methods used.

3. The formulas used have no references, are they original?

4. When presenting the results, approach the discussion in a scientific way. Don't start all chapters with Figure 1 showing ....

5. Improve the quality of your writing.

Reviewer 3 Report

This manuscript presents the results of investigations on a relevant subject matter of International Journal of Molecular Sciences. The manuscript is devoted to the study of composites of polylactic acid and corn straw fiber. Generally, the results are clearly stated in the manuscript, but there are remarks:

Comments

1). Unfortunately, the authors did not adhere to the Guide for Authors (the journal template) for preparing the manuscript.

2). Authors should check the English text carefully.

3). The method of separating CSFS and CSFC should be described.

4). The main text is written in capital letters. Check out the text of the manuscript!

5). Using the FTIR method, functional groups included in the compounds or material is determined. The structure of a material is determined by other methods.

6). For the formation of composites, the textural characteristics of the matrix are important. The textural properties of the composites must be added to the manuscript.

7). What characteristics did the authors study: absorption or adsorption? Probably, the authors do not understand the difference between these terms. What process occurs when composites are saturated with water: (I) physical adsorption; (II) capillary condensation; (III) chemisorption due to interaction with functional groups?

8). Figure 5 shows that the thermolysis of composites is two-stage: 1 - 280-300 °C and 2 - 300-320 °C. What process is associated with the second stage?

9). The term "crystallinity properties" should be replaced by "Crystallization of composites".

10). Why does the enthalpy of cold crystallization change not additively?

11). Figure 7: Have the composites been heat treated? Sample preparation conditions shall be specified.

I hope that my comments will be useful to the authors. I recommend this paper to be accepted for the publication with major revision.

Round 2

Reviewer 2 Report

I believe that the authors answered all my questions constructively. In these conditions I consider that the paper is clearer and its accuracy is improved.

I recommend accepting the paper for publication.

Reviewer 3 Report

I thank the authors for their great work in correcting the manuscript. I consider that the paper has been improved according to Reviewer’s recommendations. I recommend this paper to be accepted for the publication in International Journal of Molecular Sciences.   Best regards, Reviewer